# Comparsion of Catalyst Effectiveness in Different Chemical Depolymerization Methods of Poly(ethylene terephthalate)

**DOI:** 10.3390/molecules28176385

**Published:** 2023-08-31

**Authors:** Marcin Muszyński, Janusz Nowicki, Mateusz Zygadło, Gabiela Dudek

**Affiliations:** 1Łukasiewicz Research Network, Institute of Heavy Organic Synthesis “Blachownia”, Energetyków 9, 47-225 Kędzierzyn-Koźle, Poland; marcin.muszynski@icso.lukasiewicz.gov.pl (M.M.); janusz.nowicki@icso.lukasiewicz.gov.pl (J.N.); 2Department of Physical Chemistry and Technology of Polymers, PhD School, Silesian University of Technology, ks. M. Strzody 9, 44-100 Gliwice, Poland; 3Department of Physical Chemistry and Technology of Polymers, Faculty of Chemistry, Silesian University of Technology, ks. M. Strzody 9, 44-100 Gliwice, Poland; matezyg797@student.polsl.pl

**Keywords:** chemical recycling, depolymerization, catalysis, glycolysis, transesterification, hydrolysis

## Abstract

This paper presents an overview of the chemical recycling methods of polyethylene terephthalate (PET) described in the scientific literature in recent years. The review focused on methods of chemical recycling of PET including hydrolysis and broadly understood alcoholysis of polymer ester bonds including methanolysis, ethanolysis, glycolysis and reactions with higher alcohols. The depolymerization methods used in the literature are described, with particular emphasis on the use of homogeneous and heterogeneous catalysts and ionic liquids, as well as auxiliary substances such as solvents and cosolvents. Important process parameters such as temperature, reaction time, and pressure are compared. Detailed experimental results are presented focusing on reaction yields to allow for easy comparison of applied catalysts and for determination of the most favorable reaction conditions and methods.

## 1. Introduction

Plastics are a massively used product in the global economy. It is estimated that about 83,000 million tons of plastics have been produced since 1950 [1]. Plastics are widely used, among others, as construction materials, in the textile industry, as well as in packaging, transport and electronics [2]. The most commonly produced plastics are polyolefins, of which a significant part is poly(ethylene terephthalate) (PET), which is mainly used in the production of packaging and textiles [1]. The mass production of plastics and the lack of proper practices and regulations regarding waste collection and treatment have led to an increasing accumulation of waste in the natural environment. Untreated and poorly deposited waste enters the natural environment, leading to pollution in both land and aquatic environments. The latter has recently been a source of particular interest due to the presence of microplastics in the natural environment, which accumulates in the environment due to poor biodegradability [3,4]. The accumulation of waste and the introduction of legal regulations have resulted in an increase in interest in plastic recycling. Over the years, various methods of recycling plastics have been developed. Among these methods, energy, mechanical and chemical recycling are distinguished. Thermal recycling is used to generate energy and is used for materials that are not suitable for processing using other methods [5]. It is used for degraded plastics that can no longer be recycled, cross-linked polymers and composites whose recycling is not economical. Mechanical recycling is most commonly used for thermoplastics that can be easily recycled and reused. Chemical recycling is the process by which a polymer undergoes a chemical reaction that produces monomers or other valuable chemical compounds. Of the massively used polymers, only some can be chemically recycled. Important factors determining the possibility of chemical recycling of polymers include their susceptibility to depolymerization reaction, availability on the market and cost-effectiveness. The material most often subjected to the depolymerization process is poly(ethylene terephthalate) (PET), although there are also known methods of depolymerization of other materials such as other polyesters [6], polycarbonates [7], and polyurethanes [8]. Poly(ethylene terephthalate) is chemically recycled using such methods as hydrolysis [9], glycolysis [10] and alcoholysis [11].

A significant number of the review papers describe general issues in plastic recycling or focus on the selected polymer and describe available recycling methods in a general way. There are relatively few review publications from the last ten years describing in-depth available methods for the chemical depolymerization of poly(ethylene terephthalate) by transesterification reaction. This review includes the chemical recycling: hydrolysis, methanolysis, glycolysis, alcoholism using alcohols C_2_ to C_7_, as well as 2-ethylhexanol of PET towards useful products. These compounds can be reused in the synthesis of polymers using recovered compounds, for the synthesis of new products with practical applications or directly to new compounds with increased added value such as dioctyl terephthalate (DOTP) which can be applied as a PVC plasticizer. Furthermore, this review was carried out in terms of the types of catalysts used, divided into homogeneous catalysts, heterogeneous catalysts, as well as non-catalytic systems such as processes carried out in sub-critical and supercritical conditions. The general methodology of the conducted research and selected results obtained by the authors with an emphasis on the efficiency of the processes and the activity of catalytic systems are presented.

## 2. Chemical Recycling

### 2.1. Hydrolysis

The hydrolytic decomposition of synthetic polymers that provide starting monomers which can be reused for polymer synthesis is one of the most important methods in the chemical recycling of post-consumer polymeric materials [12]. Not every type of polymer can be processed by this method. However, polyethylene terephthalate is one of the polymers for which this processing path is possible to carry out (Figure 1).

The hydrolytic decomposition of PET for producing starting monomers was the earlier method of waste PET recycling. From a chemical point of view, the hydrolytic decomposition of PET can be carried out in neutral, alkaline and acidic conditions [13,14]. The disadvantage of this method is the relatively high cost of purifying terephthalic acid (TPA) from the post-reaction mixture, which limits the commercial use of this method for the production of high-quality polymer, e.g., intended for contact with food [14].

The neutral hydrolysis method of PET depolymerization uses water or steam without the addition of an acid or basic catalyst, usually at temperatures between 250–300 °C and pressures between 1.5–4.0 MPa. The weight ratio of PET: water is generally from 1:2 to 1:12. The neutral hydrolysis method leads directly to monomers for their subsequent use in polyester synthesis. It can be carried out both in stationary (batch) and continuous modes. In this method, various metal catalysts are often used, which positively affect the efficiency of the process, but the addition of these catalysts has a negative impact on the separation and purification of monomers, especially terephthalic acid. The literature provides a number of examples describing these types of solutions. Stanica-Ezeanu and Matei described a method of waste PET hydrolysis in neutral conditions carried out in a seawater environment [15]. The process was carried out at a temperature of 215 °C and a pressure of 4 MPa. Under these conditions, only 85–87% conversion was achieved and the TPA yield was only 76–84%.

Under the standard hydrothermal process (250 °C, 39–40 bar, 30 min.), 90–92% PET conversion can be achieved [16,17]. Depolymerization of the colored polymer is slightly less effective. Conversion of PET obtained in this case was only 85% [17]. The process can also be carried out in supercritical conditions (H_2_O or CO_2_) [17] and also supported by microwave heating [18,19]. Although this method can be considered as effective and more ecological, it requires the use of more drastic conditions of temperature, pressure and reaction time, which can be up to 5 h.

Based on recent literature, methods of PET depolymerization (hydrolysis) carried out in alkaline conditions, most often using NaOH as an alkaline catalyst, are definitely dominant [20,21,22,23]. Both the PET conversion and TPA yields obtained were typically >90%. There are also known solutions in which the hydrolysis process is carried out in a mixture containing an additional solvent, such as ethanol [24] or γ-valerolactone used as pre-solvent of waste PET [25]. Specific phase transfer catalysts were also used as catalysts supporting the hydrolysis process [26,27,28]. They turned out to be very effective in the tested reaction, as evidenced by the very high PET conversion values (99%) obtained. Both classical ammonium salts and highly specific ammonium phosphotungstates were used as the phase transfer catalyst (Table 1).

In alkaline hydrolysis, separation of terephthalic acid requires an additional precipitation operation using acid solutions, which generates an additional stream of waste inorganic salts. These problems do not occur in hydrolysis processes carried out under acidic conditions. As catalysts, both strong sulfonic acids, such as H_2_SO_4_ [28] or *p*-toluenesulphonic acid, were commonly applied [29]. Some heterogeneous catalysts with acidic properties, such as heteropolyacids [28] and “superacid” type SO_4_^2−^/TiO_2_ [30] and WO_3_/SiO_2_ [31] were also used as catalysts in PET hydrolysis process. In the case of sulfuric acid, a PET conversion of 83% was obtained with a TPA yield of about 75%. Similar results were obtained for heteropolyacids; however, superior results were obtained for ptc-type quaternary ammonium phosphotungstates, for which PET conversion was even 100% [28]. Yang et al. described an interesting process of PET hydrolysis catalyzed by easily recyclable terephthalic acid [32]. The advantage of this method is that no compounds are introduced into the process that would require removal from the post-synthesis mixture. PET conversion was close to 100% with a TPA yield of 95.5% (Table 2).

### 2.2. PET Alcoholysis

Alcoholysis of poly(ethylene terephthalate) (Figure 2) is, apart from alkaline hydrolysis, one of the basic methods of chemical depolymerization of waste PET. Alcoholysis involves the degradation of PET in an alcohol environment under conditions of high temperature and pressure. Alcoholysis is considered to be one of the more reliable and effective methods applied for waste PET recycling [33,34]. Alcoholysis processes use a wide group of well-known alcohols, among which the most often used are: methanol, ethanol, butanol and isooctyl alcohol (2-ethylhexanol). Among the described reports in this field, various variants of methanolysis processes definitely dominate, which is related to the key role of dimethyl terephthalate obtained in these processes. As it is known, dimethyl terephthalate is the basic raw material in the synthesis technologies of phthalate polyesters. Alcoholysis using alcohols > C_1_ has become less common and esters of terephthalic acid and higher alcohols are often used as raw materials in processes other than the synthesis of polyesters.

#### 2.2.1. Methanolysis

Methanolysis process is used to produce dimethylterephthalate (DMT) which then can be applied in poly(ethylene terephthalate). DMT is produced via the reaction of PET with methyl alcohol usually under increased pressure due to the low boiling point of methanol. Reaction is conducted in the presence of various types of catalysts such as Bronsted and Lewis acids, hydroxides, organic bases, oxides and ionic liquids. The process is most often conducted in liquid phase; however, gas-phase methanolysis [35] and processes conducted in supercritical conditions are described in the scientific literature [36]. Separation and repeated usage of catalysts from depolymerization products is an important aspect of designing a viable process that can be implemented as a working chemical technology. Homogenous catalysts can be difficult to remove and to reuse. To remove this obstacle, the use of heterogeneous catalysts in chemical depolymerization of PET is investigated.

One of the catalysts in depolymerization reaction of poly(ethylene terephthalate) is zinc acetate. Its application as a PET depolymerization catalyst was described in numerous scientific publications and is characterized by high activity and its application results in high product yields. Hofmann et al. [37] investigated methanolysis of waste PET using zinc acetate catalyst and waste PET. The process was conducted in the presence of dichloromethane which acted as a solvent. The authors obtained high yields of dimethyl terephthalate which reached 98% after a 20 min time period. However, a large amount of methanol was used in this process. Equivalent methanol-to-PET repeating unit ratio varied from 46.2 to 92.5. Catalyst amount was weight in respect to amount of PET used in the reaction. Interestingly, reaction yield dropped significantly after lowering temperature to 140 °C. In this case, after 20 min reaction yield was under 1%wt and reached 92%wt after 60 min of reaction. Employed catalyst is also susceptible to contaminants in waste PET. When reaction was carried out using dyed bottles, this resulted in lower reaction yield of 38%wt. Investigation of thermodynamics and kinetics of waste PET methanolysis was conducted by Mishra et al. [38] using zinc and lead acetates as catalysts. Influence of PET particle size on reaction kinetics was investigated and reaction was also optimized. Optimal reaction time was determined to be 120 min at temperatures ranging from 130 to 140 °C with PET particle size of 127.5 μm. Zinc acetate was also employed as a catalyst in chemical recycling of mixture of waste PET and polylactic acid (PLA) [37] using methanol as well as a number of other solvents, i.e., ethanol, ethylene glycol, etc. After reaction conducted in 15 h time under boiling point of methanol, there was no discernable effect on PET while all of the PLA was depolymerized. This result is attributed to differences between solubility of tested polymers in methanol. Methanolysis of PET in microwave reactor was investigated in [6] and the process is characterized by a very short reaction time and low amount of catalyst used. Within 10 min of the process with catalyst loading of 0.01 g per 1 g of PET, 88%wt of PET can be converted.

Some inorganic as well as organic catalysts are active at low temperatures in methanolysis of waste poly(ethylene terephthalate). Catalysts like potassium carbonate 1,5,7-triazabicyclo[4.4.0]dec-5-ene (TBD) and potassium methoxide CH_3_OK exhibit good catalytic properties in methanolysis reaction conducted in low temperatures [39]. Methanolysis conducted at a temperature of 25 °C over a period of 24 h using K_2_CO_3_ yielded 93.1%wt dimethyl terephthalate. To obtain such high yields, a large excess of methanol and dichloromethane was used as well as large amounts of catalyst. Implementation of TBD and CH_3_OK resulted in lower yields of product 89.3 and 85.5%wt, respectively. Interestingly, K_2_CO_3_ exhibits highest catalytic activity although it did not completely dissolve in applied solvent. Other catalysts like KHCO_3_, KOAc, Na_2_CO_3_, and CaO among others had given significantly lower concentrations of dimethyl terephthalate.

Calcinated sodium silicate (Na_2_SiO_3_) was used in the methanolysis of PET by Tang et al. [40]. The authors investigated catalyst obtained by calcination under different temperatures as well as influence of catalyst concentration, reaction temperature methanol-to-PET weight ratio and reaction time on reaction yield and PET conversion. Applied silicate exhibits good catalytic properties in relatively small catalyst loadings from 3 to 7%wt reaching up to 63% yield and 74% conversion rate using as much as 5% of the catalyst. The process was tested in temperatures ranging from 160 to 200 °C. Interestingly, the authors used relatively low alcohol-to-PET ratio which ranged from 3 to 7. Process conducted under optimized conditions resulted in obtaining dimethyl terephthalate with 95% yield and 100% conversion. Recycling of catalyst was also investigated. Silicate catalyst was reused four times with some loss in activity attributed to adsorption of water. Magnesium phosphate catalyst obtained in the presence of pectin was used in PET methanolysis [41]. The use of pectin resulted in obtaining catalyst with large BET surface area of 19.51 m^2^/g and average pore size of 26.01, which was significantly higher compared to catalyst obtained without the use of pectin. Process was conducted at 180 °C for 150 min with 3%wt of catalyst achieving yield of 74%wt. MgP catalyst is stable when reused as it was reused four times with low loss in PET conversion. However, a methanol-to-PET weight ratio of 200 was used in this study. Such large alcohol excess will have a negative influence on the overall amount of obtained DMT per synthesis.

Heterogeneous catalysts obtained from bio wastes can also be implemented in methanolysis of waste poly(ethylene terephthalate) [42]. The authors used a catalyst obtained by calcination of bamboo leaf at 700 °C in methanolysis of PET waste. The obtained catalyst was composed mainly of SiO_2_ and a mixture of various oxides of other metals such as calcium, potassium, iron, manganese, magnesium, etc. Methanolysis reaction using such catalyst allowed for achieving DMT with a yield of 78%wt after two hours in relatively low methanol excess of 7.5 and catalyst loading of 20.8%wt in relation to the mass of PET. Interestingly, increase in catalyst loading, reaction temperature and time resulted in lower yield of DMT. Reusability tests have shown the loss in activity of the catalyst. After four cycles, DMT yield lowered from 78%wt to 67%wt. Nanocatalysts in the form of zinc oxide dispersions are found to be active in methanolysis of poly(ethylene terephthalate) [43]. Depolymerization process conditions were optimized regarding reaction time, methanol-to-PET ratio and catalyst concentration. The tested catalyst exhibits very good activity achieving DMT yield of 97%wt after 15 min at 170 °C and subsequent trials conducted to test the possibility of catalyst reuse have shown a decrease in activity by approximately 20%. Overall, ZnO nanodispersion has proven to be an active catalytic system which allows for obtaining high yields in very short time periods. Heterogeneous hydrotalcite (Mg-Al) has proven to be an effective degradation catalyst when used in conjunction with dimethyl sulfoxide (DMSO) [44]. Degradation process was completed in a 10 min time period obtaining PET oligomer. Obtained product was then reacted with methanol in the presence of sodium hydroxide (NaOH) at 35 °C for 60 min.

Ionic liquids are used as methanolysis catalysts in depolymerization of poly(ethylene terephthalate). Liu et al. [45] tested a series of ionic liquids in methanolysis of various polymers including poly(ethylene terephthalate), polycarbonate, Polyhydroxybutyrate and polylactic acid. [HDBU][Im] and [Bmim]_2_[CoCl_4_] were used in PET methanolysis. Reaction was conducted at 170 °C over a period of four hours for [Bmim]_2_[CoCl_4_] and 140 °C over a period of three hours for [HDBU][Im]. Reactions yielded 78 and 75%wt, respectively.

Chemical recycling of waste poly(ethylene terephthalate) under supercritical conditions can be employed successfully for PET methanolysis [46]. Process conducted at 298 °C for a duration of 112 min with excess methanol results in DMT yield of 99.79%wt (Table 3).

#### 2.2.2. Glycolysis

Glycolysis is a widely used method for the chemical processing of PET. Most commonly, as the reactive solvent, ethylene glycol (EG) and diethylene glycol (DEG) are used [14]. From a chemical point of view, the glycolysis reaction is the decomposition of PET with glycols, in the presence of transesterification catalysts, where ester linkages break and are replaced with hydroxyl terminals. One of the earliest patents of this process was proposed in 1964 by Ostrysz et al. and concerns the preparation of unsaturated polyester resins (UPR) bearing a structure as presented in Figure 3 [49,50].

Glycolysis proceeds in a several steps as presented in Figure 4. Initially, PET is converted to its dimer form through oligomer molecules at a high rate, and then slow transition takes place. Process is mainly carried out in mild conditions under atmospheric pressure and at a temperature ranging between 180–240 °C. A major advantage of this procedure is the application of less volatile glycols compared to alcohols of similar structure [51,52]. It was also found that specific order of parameters: catalyst concentration > glycolysis temperature > glycolysis time affects the yield of obtained product [53,54]. However, the disadvantage of this process is that the yield is insufficient for industrial application. In order to increase efficiency of the process, catalysts are used [5,55]. The most widely discussed group of catalysts in the recent literature are salts of transition group metals. Zinc (II) Acetate (Zn(OAc)_2_) had been found to be the most effective catalyst. Zn(OAc)_2_ was applied for the first time as a catalyst for PET glycolysis in 1989 by Vaidya and Nadkarni; however, other molecules have been topics of interest in recent studies. These are mainly transition metals salts and complexes, ionic liquids and metal oxides [5,56].

Processed PET in the form of polyhydric alcohols or oligomers with applied glycol on the end-groups have found numerous applications. Depolymerized PET can be used as an additive to poly(vinyl chloride) (PVC) [5]. Another application of oligoesters is in the production of polyurethanes, as polyol groups containing reactant. The depolymerized PET with ethylene glycol can be reused to produce fresh poly(ethylene terephthalate) molecules [57].

Metals salts and complex compounds

Zn(OAc)_2_ was one of the first catalysts applied in the depolymerization process of poly(ethylene terephthalate). Wang et al. [58] glycolyzed PET fibers with EG at a weight ratio of 1:3 and Zn(OAc)_2_ as the catalyst. The reaction was carried out at 196 °C under nitrogen atmosphere at standard pressure for 0.5 h for differing catalyst concentrations ranging from 0 to 0.8%wt of PET load. They found that in absence of the catalyst, the reaction almost fails to proceed reaching only 2.79% PET conversion, while with the use of a catalyst at 0.2% concentration, PET conversion exceeded 93%. The highest productivity was obtained at 196 °C with 0.2% of Zn(OAc)_2_ and PET/EG ratio 1:3 in 2 h, reaching 100% PET conversion and an 81.8% yield of bis(hydroxyethyl) terephthalate (BHET). Song et al. [59] analyzed the process of PET glycolysis with EG applying tropine and tropine-Zn(OAc)_2_ complex as catalysts. Process was conducted at atmospheric pressure at five temperatures between 150 °C and 190 °C with 10 °C steps. The most effective conditions were established at 170 °C with 5%wt of the catalyst with respect to PET and was achieved over the course of 2 h. The final yields of BHET were then found to be 91% and 182% higher for tropine and tropine-Zn(OAc)_2_ complex, respectively, than for Zn(OAc)_2_ reaching only about 15%. Moncada et al. [60] studied progress of PET glycolysis at different temperatures 165 °C, 175 °C and 185 °C, with Zn(OAc)_2_ catalyst at a concentration of 1%wt and 0.5%wt of initial mass of PET and in presence of EG (2.4 mL along with 1 g of PET). Reaction progress was established by measuring water-insoluble mass fraction of depolymerized PET along 80 min in the oil bath, with 10 min sampling intervals. They found that the highest depolymerization progress is observed after first 30 min at 185 °C reaching about 13% of initial average molecular weight while at 165 °C it was only about 35%. Similar results were found for both catalyst concentrations. PET with decreased molecular weight was then used for block co-polymer building, consisting of PET oligomers and poly(ethylene glycol) [61].

Garcia et al. [62] conducted a series of experiments on metal salts and complexes as catalysts for PET glycolysis with ethylene glycol. Applying different concentration of catalysts and reaction conditions, they attempted to find a substance that would effectively catalyze the PET glycolysis process. The results of their work are presented in Table 4.

As it can be seen, the most efficient catalyst for the PET glycolysis process is a complex compound of cobalt dichloride with 1,2-Bis(dicyclohexylphosphino)ethane ([Co(dcype)Cl_2_]). They also pointed out a reverse correlation between the amount of ligand added and reaction yield. After increasing the number of dcype, the reaction yield decreased. Furthermore, the efficiency of the process depends on the applied ligands and for catalytic purposes the bidentate ligand should be used, while monodentate inhibits the process of PET glycolysis [62].

Liu et al. [63] chose another approach for reaction yield improvement. They decided to conduct the glycolysis reaction in a homogenous environment. For this purpose, they applied dimethyl sulfoxide (DMSO) as a solvent for PET. They conducted the glycolysis reaction in two variants. First, 5 g of PET powder was mixed with 30 g of EG and heated to temperatures ranging from 160 °C to 190 °C for 5 to 300 min. Second, 5 g of PET powder was mixed with 10 g of EG and 20 g of solvent, then heated in temperature range from 155 °C to 190 °C for 1 to 20 min. For both cases, the catalyst was added when the assumed temperature was reached. Results of their experiments from HPLC analysis are presented in Table 5.

The obtained results directly indicate a high positive correlation between solvent application and BHET yield. With each catalyst, the yield increased by a significant amount. Moreover, the reaction time is extremely small. In most cases, lengthy time periods in the order of hours are required to achieve a BHET yield of about 80%, while the addition of solvent reduces this time to minutes. This has a great impact on energy efficiency of the process. However, the process of applying organic solvents is not environmentally friendly and should be avoided in order to reduce the impact on living organisms. It is for this reason that such a process might be considered economically suitable, but environmentally harmful [63].

Lei et al. [60] applied tin (II) chloride (SnCl_2_) as a catalyst in glycolysis process. They conducted studies on increasing glycolysis efficiency by stepwise PET addition process. Process was carried out in diethylene glycol (DEG) at 200 °C, 210 °C, and 220 °C. Weight ratios of PET-to-DEG were 1:2, 1:1.5, and 1:1 accompanied with tin (II) chloride (SnCl_2_) 0.3%wt of loaded PET for up to 150 min. In the one-step process, all reactants were mixed together and heated to the desired temperature. For the stepwise process, DEG was mixed with SnCl_2_ and heated, then the PET had been added in small portions with regular intervals. Time measuring was started when the last portion was added. The reaction yield (WSM%) was established as an amount of water soluble monomer (WSM) in ratio with a theoretical amount of water soluble monomer. The highest efficiency was obtained at 220 °C after 90 min of the reaction. Presented results exhibit high improvement on the reaction yield for stepwise process, reaching over twice as much WSM% as the one-step addition. Moreover, higher WSM% was obtained for PET waste bottles and purchased particles than for PET fibers. Authors explained that phenomenon with higher crystallinity of PET fibers, that impedes the penetration of the degrading agent. They also conclude that the reaction takes place mostly at the solid/liquid interface of PET and DEG, though stepwise PET inserting allows for more rapid swelling and dissolving of solids. This has a significant influence on the reaction efficiency and explains the difference between single and multiple step procedures.

Metal oxides

In a study by Son et al. [64] on PET glycolysis, the authors applied exfoliated nanosheets of manganese dioxide (e-MON) as an improvement of the process conducted with typically used MnO_2_ forms. e-MON was obtained from δ–MnO_2_ in a fluid dynamic reactor in Taylor–Couette flow, and then separated with a centrifuge. With this method, they obtained MnO_2_ sheets with a few nanometers thickness and relatively large surface area, what has a crucial influence on the catalyst efficiency. Glycolysis was carried out with an EG in mass ratios EG/PET in range from 6.7 up to 55.5. Process temperatures were controlled from 150 °C to 200 °C. An amount of the e-MON was kept constant 0.01%wt of PET. Comparison experiments with bulk MnO_2_ were also conducted. The results clearly show that e-MON is a much more efficient catalyst than bulk MnO_2_, reaching 100% yield of BHET synthesis in 30 min at 200 °C, while reaction with bulk MnO_2_ reached only about 80%. The highest BHET yields were obtained for EG/PET mass ratio starting from 18.5 and higher reaching 100%. After the process, e-MON was recovered by washing with deionized water and filtration. Recovered e-MON was then used four-more times without losing its catalytic properties still reaching 100% BHET yield synthesis.

Wang et al. [65] used as a PET glycolysis catalysts CoFe_2_O_4_ magnetic nanoparticles (MNP) modified with ionic liquid as a surfactant. Modified NMP’s were synthesized by precipitation from aqueous solution of iron and cobalt salts with addition of certain ionic liquid. Modified NMP’s were crushed out of the solution through the addition of acetone and separated using magnets. Size distribution of modified MNP’s depends on the applied ionic liquid ranging from 4 nm to below 20 nm, while not “coated” MNP has a size distribution range from 20 nm up to 35 nm. For the glycolysis reaction, they used 5 g of PET and 0.1 g of the catalyst. The most efficient amount of EG was distinguished as 25 g within the range 10–30 g, while the reaction time with the highest BHET synthesis yield was found as 2.5 h within the 1–3.5 h range. The results of their work are presented in Table 6.

The results in Table 6 clearly indicate that from the tested catalysts, the most efficient is MNP coated with ionic liquid with a formula [C_10_COOHbim]OAc reaching a BHET synthesis yield of 95.4%. A huge advantage of using modified MNP’s is the possibility to easily separate them from the post-reaction mixture by applying a magnetic field. Separating this way, CoFe_2_O_4_/[C_10_COOHbim]OAc was used over six glycolysis cycles without impacting PET conversion and BHET yield. From seventh to tenth attempts, BHET yields slightly decreased to about 95% [63].

Guo et al. [66] introduced micro iron oxide particles coated with magnesium aluminum oxide (Mg-Al-O@Fe_3_O_4_) to the PET glycolysis reaction. Mg-Al-O@Fe_3_O_4_ was prepared from two solutions. Solution A contains sodium hydroxide and sodium carbonate and solution B consists of iron oxide suspension, magnesium nitrate and aluminum nitrate. Solution A was dropwise added into the solution B until pH of the mixture reached 10, then it was left for aging. After 24 h, the mixture was freeze dried and annealed at 1000 °C. Glycolysis reaction was conducted using 5 g of PET along with 25 g of EG. Analyzed amounts of the catalyst were 0.1, 0.25, 0.5, 1.0 and 2.0%wt of PET. Reaction was carried out at different temperatures 200 °C, 220 °C, 240 °C and 250 °C. The reaction time ranged from 20 to 150 min.

They found that the most efficient glycolysis parameters are 240 °C for 90 min with the catalyst concentration 0.5%wt of PET. Under such conditions, BHET synthesis yield reaches 82%mol. The great advantage of such catalyst is its easy recovery through application of an external magnetic field. The catalyst can be used in a cycle. For the second reaction, BHET yield dropped only slightly; however, the third run reached only about a half of the initial yield. For the fourth cycle, the catalyst becomes fully deactivated obtaining less than 10%mol of BHET yield. Fortunately, catalyst regeneration by annealing at 1000 °C recovers most of the catalyst efficiency reaching nearly the same yield as in the second run.

Fang et al. [67] studied the catalytic properties of sandwich-structure polyoxometalates (POMs) for PET glycolysis. They analyzed molecules with the formula Na_12_[WZnM_2_(H_2_O)_2_(ZnW_9_O_34_)_2_] where M = Zn^2+^, Mn^2+^, Co^2+^, Cu^2+^ and Ni^2+^. For the reaction, 5 g of PET was mixed with 20 g of EG in a reactor. PET-to-catalyst molar ratio was 0.018%mol, the temperature was kept at 190 °C and the time duration ranged from 40 min to 110 min. Experimental results are given in Table 7.

They found that the most efficient POM was Na_12_[WZnCo_2_(H_2_O)_2_(ZnW_9_O_34_)_2_] reaching a BHET yield of 84.61% in only 40 min. The authors also state that POMs are relatively cheap and with low toxicity catalysts. They also recover the remaining catalyst from the post-reaction mixture and use it in a cycle. Within four cycles, the catalyst kept nearly the same BHET yield confirming its stability as declared by the authors.

Putisompon et al. [68] in their research found that calcium oxide (CaO) has applications as a catalyst for transesterification in biodiesel production and decided to test its potential to catalyze PET glycolysis process. For this purpose, as a source of CaO ostrich eggshells, chicken eggshells, mussel shells, geloina shells, and oyster shells from Thailand local flesh market were chosen and Zn(OAc)_2_ as a reference catalyst. Shells were washed, ground, and calcined in variety of temperatures from 600 °C to 1000 °C with 100 °C step. Calcined shells were then ground to a fine powder. To the reactor, 5 g of PET was loaded along with 1%wt of a catalyst. Weight ratio of PET to EG 1:15 was applied. The reaction was carried out at 192 °C for 2 h. Then, obtained BHET was extracted with water, filtered out, dried, and weight to establish the final yield as a percent of BHET.

The most efficient calcination temperature of 1000 °C was established, reaching 72 and 76% of BHET for a chicken eggshell and an ostrich eggshell, respectively. These results increased from barely several percent at a calcination temperature equal to 600 °C. When the decomposition of calcium carbonate to CaO starts at 700 °C, temperature increase also reduces organic impurities present in the shell structures and increase crystallinity of obtained CaO [65].

Zangana et al. [69] covered the topic of microwave irradiation support for the catalytic glycolysis of PET. They conducted the process using different molar ratios of PET:EG such as 1:4, 1:6, 1:10 and 1:20 with reaction times up to 5 min. As a catalyst, CaO was applied in concentrations 3%wt, 5%wt and 10%wt of the reaction mixture (PET and EG). Reaction was carried out at EG boiling temperature 197 °C, supported by microwave irradiation with power 800 W. For optimized conditions, supported CaO on activated carbon (CaO/AC) as a catalyst was applied. For the CaO, the most optimal reaction conditions are 10%wt of the catalyst, PET:EG ratio 1:10 and 4 min of microwave irradiation reaching about 75% BHET yield, while for CaO/AC 15%wt of catalyst along 3.5 min of microwave irradiation with the same PET:EG ratio similar yield was obtained. They also analyzed catalyst and EG recovery as well as their re-use over 3 cycles. For each run, CaO concentration was kept constant; however, the reaction time was increased for each cycle reaching 16 min on third run. Catalyst efficiency decreased on each run, but what is interesting is that CaO/AC shows much smaller differences than CaO. Recovered CaO and EG behaves at the second run as recovered CaO [66]. It is worth mentioning that Zangana et al. [69] and Putisompon et al. [68] obtained similar BHET yields; however, Zananga et al. [69] reached the final concentration about 30-times faster due to microwave support.

For the glycolysis process, Lalhmangaihzuala et al. [70] applied orange peels ash (OPA) as the catalyst. Orange peels were initially washed with distilled water and dried at atmospheric conditions or in an oven. In the next step, dry peels were burned. Glycolysis reaction was conducted with 480 mg of PET with certain amount of EG and OPA. Reactor was placed in an oil bath and heated to 190 °C until complete disappearance of PET. Reaction parameters and results are presented in Table 8.

As it can be seen, the most efficient parameters of PET (480 mg) glycolysis with OPA at 190 °C were a reaction time set as 1.5 h, 2.5 g mass of EG and 50 mg of OPA, which produces 521%wt (of PET) EG and 10.42%wt (of PET) OPA [67]. In conclusion, it is possible to apply waste orange peels as an efficient catalyst for the glycolysis reaction of a waste PET.

Ionic liquids

Shuangjun et al. [71] investigated Lewis acidic ionic liquids (LAIL) as a possible active catalytic system for the glycolysis process of the post consumption PET. For this purpose, LAIL’s consist of 1-hexyl-3-methylimidazolium (Hmim) and metal halides (MH) (zinc, cobalt, iron and copper chlorides) were prepared in different molar ratios of Hmim and MH, and mixtures of two LAIL’s were also analyzed. To conduct glycolysis reaction, 2 g of PET was loaded into the reactor along with 22 g of EG and 0.1%wt (of PET) of the catalyst. The reaction was carried out for 2 h at 190 °C. Results of these processes are presented in Table 9.

Among all the catalytic compositions, the most efficient composition reached 87.1 BHET yield along with 100% PET conversion, which was [Hmim]ZnCl_3_:[Hmim]CoCl_3_ at a 1:1 molar ratio. What is also interesting is that pure components [Hmim]Cl and MH exhibit lower BHET yields than when mixed forming LAIL’s. Moreover, systems with pure [Hmim]CuCl_2_ and mixture of [Hmim]ZnCl_2_ and [Hmim]CuCl_2_ in molar ratio 1:3, respectively have very low PET conversion along with no BHET detected. Basing on their results, authors purposed a decreasing order of metal anions activity: [ZnCl_3_]^−^ > [CoCl_3_]^−^ > [FeCl_4_]^−^ > [CuCl_3_]^−^ [71].

Cano et al. [72] applied MNP consists of the core made of Fe_3_O_4_ coated with SiO_2_ and ionic liquid (IL) as a catalyst. General procedure for PET glycolysis was done by mixing 100 mg of PET and 1 mL of EG in the reactor with 15 mg of MNP. The reaction was preceded for 24 h at 160 °C or 180 °C. They also recovered MNP using an external magnetic field, then applied the same catalyst portion in over a dozen times. At 160 °C reaction yields of BHET were rather low reaching less than 60% with a maximal PET conversion about 64% dropping into poor yield about 20% after fifteenth cycle. In contrast, reaction conducted at 180 °C reached 100% yield after first cycle, while for the next 10 repetitions yield was kept above 90%. In the last twelfth step yield dropped to the still significant yield 84%. All twelve cycles carried out at 180 °C exhibit 100% PET conversion.

Najafi-Shoa et al. [73] in their studies on PET glycolysis applied IL grafted on graphene. For that purpose 1-(triethoxysilyl) propyl-3-methylimidazolium chloride ([TESPMI]Cl) is attached to the graphene oxide (GO), which is then reduced with hydrazine forming rGO/[TESPMI]Cl. The final catalyst is prepared by dispersion of rGO/[TESPMI]Cl at CoCl_2_ solution, what leads to formation of rGO/[TESPMI]_2_CoCl_4_. They have done the glycolysis reaction by mixing 1 g of PET with 8 to 14 g of EG along with 0.05 up to 0.25 g of catalyst. The reaction was carried out from 1 to 4 h at the temperature in range from 150 °C up to 190 °C. What they found is the most efficient conditions are for the process with 14 g of EG proceed for 3 h at 190 °C with 0.15 g of a catalyst. That allows to reach 95.22% BHET yield with 100% PET conversion. Authors also verified the reusability of the catalyst. What they obtained, is that after fifth cycle BHET yield drops to about 90%, while the PET conversion remains unchanged at 100% level.

Others catalytic systems

Hong Le et al. [74] noticed that most of the glycolysis processes require high energy consumption, which is mostly allocated to heating of the reaction system. They have decided to apply anisole as co-solvent for reduction of an energy requirements of the process. Glycolysis was carried out for 10 g of PET with 38.75 g of EG and 22.51 g of anisole. Reaction had been performed for two hours. The variety of catalyst had been tested, such as Zn(OAc)_2_, NaOAc, KOAc, Na_2_CO_3_, K_2_CO_3_, NaHCO_3_, KHCO_3_, MgCO_3_, potassium methoxide (CH_3_OK), 1,1-DMU, 1,3-DMU and 1,5,7-triazabicyclo [4.4.0]dec-5-ene (TBD). Comparing results for the reaction done at 197 °C without co-solvent and at 153 °C with anisole with molar ratios EG:anisole:catalyst:PET equal to 12:4:0.04:1 KOAc was depicted as the most efficient catalyst reaching (at 153 °C with anisole) about 87% BHET yield with 100% PET conversion, while reaction at 197 °C without co-solvent with KOAc reached about 84% BHET yield with the same PET conversion. Nevertheless, reaction carried out at 153 °C with the molar ratios of EG:anisole:catalyst:PET equal to 10:3:0.02:1 only TBD got BHET yield above 80% and 100% PET conversion. Other co-solvents were also tested, however like in the previous sentence, only for anisole 80% BHET yield boundary was crossed reaching also 100% PET conversion.

Veregue et al. [75] in their research work on PET glycolysis involved cobalt nanoparticles (CoNP) too check its activity as a catalytic system. For that purpose CoNP were synthesized through the reduction reaction of cobalt (II) chloride with sodium borohydride. The depolymerization reaction was achieved for 4 g of PET with 100 mL of EG and 60 mg of CoNP. The reaction was carried out at 180 °C for 2, 3 or 4 h. They have reached maximum BHET yield with the time of the process equal to 3 h reaching 77%. Catalyst was then recovered from the postreaction mixture and used four more times to check its activity. They found that CoNP retained its catalytic properties for all five cycles.

In contrary to the most of research done on the PET glycolysis, Wang et al. [76] have decided to check organocatalyst cyanamide efficiency for this process and find optimal reaction conditions. In their procedure, they mixed 2 g of PET with 20 g of EG. They have analyzed reaction yield in terms of temperature in a range between 160 °C to 200 °C, time in a range 0.5 h up to 3 h and catalyst weight percent with respect to PET in a range from 2.5% to 15%. They obtained the most efficient reaction conditions at the temperature 190 °C in 2.5 h with 5%wt of cyanamide. With these parameters they have reached 100% PET conversion along with about 95% BHET yield. What remains interesting, further increase of mentioned parameters was inversely related with BHET yield, which started to decrease, however PET conversion was kept at 100% level.

Wang et al. [77] checked catalytic activity of graphite carbon nitride colloid (GCNC) in the PET glycolysis process. Moreover, they attempt to establish the most efficient parameters for this reaction. For that purpose they have mixed 2 g of PET with 20 g of EG. Then following reaction conditions were investigated: temperature in a range from 160 °C to 196 °C; time in a range from 5 to 120 min; GCNC mass in a range from 0.01 g to 0.15 g. They found optimal parameters for PET glycolysis with GCNC as 196 °C for 0.5 h along with 0.5 g of the catalyst. In such conditions 80.3% of BHET yield had been co-established with 100% PET conversion.

#### 2.2.3. Ethanolysis

Ethanolysis of poly(ethylene terephthalate) is widely regarded as an environmentally friendly alternative to methanolysis, which has been well known for many years. This is mainly due to the possibility of replacing harmful methanol with safer ethanol [78]. Diethyl terephthalate formed in the depolymerization (hydrolysis) reaction can be reused for the synthesis of poly(ethylene terephthalate) (Figure 5).

The use of ethanol in the process of depolymerization of waste PET should be considered in a much broader aspect than only as a process leading to diethyl terephthalate. Speaking about the process of ethanolysis of poly(ethylene terephthalate), one should also take into account the process whose final product is terephthalic acid and it is a process similar to the process of alkaline PET hydrolysis described earlier. Three reactions take place under alkaline catalysis (NaOH) in a water-ethanol medium: depolymerization of PET to a mixture of diethyl terephthalate (Step I), its hydrolysis to TPA disodium salt (Step II) and precipitation of free TPA (Step III, Figure 6). Ethanol is recycled and can be reused (Closed-loop recycling process), making it very competitive with conventional alkaline hydrolysis processes [24,79,80].

The process can be carried out under relatively mild conditions (reaction temperature 50–80 °C, the proportion of ethanol to water 20–100 vol%, the amount of NaOH 5–15%wt) [81]. Under optimal conditions, TPA efficiency reaches 95%. In the method presented in Figure 6, diethyl terephthalate formed in Step I is not separated. Under process conditions, it hydrolyzes to terephthalic acid.

Diethyl terephthalate is obtained by direct ethanolysis of PET (Figure 5). The direct PET ethanolysis process requires temperatures above 200 °C and sometimes the addition of a catalyst. Li et al. [82] described a method of direct PET ethanolysis carried out under pressure and at a temperature of 180–200 °C without the addition of a catalyst. The author reported that under the assumed reaction conditions, nearly 100% PET conversion can be obtained and the DET yield reached 97%. Zinc acetate is an effective catalyst for PET ethanolysis, which allows DET yields of 96–97% to be achieved [83,84]. From a technological point of view, the method is very convenient. Due to differences in solubility, the ethylene glycol formed in the reaction is the lower, immiscible phase, which can be separated from the upper phase containing diethyl terephthalate. Compounds containing acidic sulfonic groups, such as, sulfonic ionic liquids, can also act as catalysts for the PET ethanolysis reaction [85]. By carrying out the process for 14 h at 80 °C in the presence of sulfobutylammonium ionic liquid, diethyl terephthalate can be obtained with a yield of 96%. However, using the techniques of depolymerization of waste PET leading to the production of useful phthalate monomers (terephthalic acid esters), solvolysis processes carried out in supercritical conditions prevail. The basics of the process were presented in the last century in Japan, where a method of PET depolymerization using supercritical water and methanol was developed [78]. Supercritical solvents are very attractive media for conducting many chemical processes mainly because the solvent and transport properties of a single solution can be appreciably and continuously varied with relatively minor changes in either temperature or pressure. Variation in the supercritical fluid density also influences the chemical potential of solutes, reaction rate and equilibrium constant [86]. Depolymerization of waste PET in supercritical methanol is the subject of extensive research [87]. In recent years, however, there have been increasing reports on solvolysis of waste polyester polymers in supercritical ethanol [78,88]. As described by Castro et al. [78], the ethanolysis process was carried out in a supercritical ethanol environment at a temperature of 255 °C, a pressure of 115 or 165 bar and reaction times between 5.0 and 6.5 h without the addition of a catalyst. Under these process conditions, PET was practically completely depolymerized and the main product of the reaction, apart from ethylene glycol and ethanol, was diethyl terephthalate. Later studies included the addition of catalysts, such as metal oxides [89,90] and ionic liquids [91,92]. As the research has shown, the addition of catalysts to the PET ethanolysis process does not have a major impact on the conversion rates, but conclusively shortens the reaction time. Later studies showed that increasing the temperature to 275–350 °C removed the need for the addition of catalysts while maintaining high PET conversion rates and DET efficiency [93,94]. Recent studies have also shown the high efficiency of the PET ethanolysis process in supercritical conditions and the possibility of using this method on a technical scale (Table 10).

#### 2.2.4. Alcoholysis with Alcohols > C_2_

Diethyl terephthalate is used as a safer alternative to dimethyl terephthalate in the synthesis of polyesters. Terephthalates of higher alcohols are no longer used in this specific application. However, they are equally valuable due to their plasticizing properties. This also applies to dibutyl terephthalate, which is produced on an industrial scale. Dibutyl terephthalate has fast melting and low migration properties and provides greater flexibility of finished products. These properties mean that it is used, for example, in the production of flexible floor coverings based on PVC, adhesives and sealants, and in the production of printing inks [95,96].

Dibutyl terephthalate is most often produced by esterification of terephthalic acid, but the use of waste for its synthesis seems to be a more ecological alternative. Depolymerization of waste PET by butanol alcoholysis requires a completely different process than in the case of alcohols such as methanol or ethanol. It also requires the addition of a catalyst. According to the description given in the patent CN102603532, the PET depolymerization process in the presence of 1-butanol and sulfonic ionic liquid allows dibutyl terephthalate to be obtained with a yield of 93.9%. According to the description, the alcoholysis process was carried out at 190 °C for 9 h [85]. In turn, the patent description WO2022112715 provides a general method for the synthesis of terephthalic acid esters, including dibutyl ester [97]. In this method, waste PET is reacted with excess butanol at a temperature between 50–70 °C for 1.5–3 h in the presence of a catalyst being a suitably selected mixture of cyclic guanidine (TBD) or amidine (DBU) derivatives and sodium methoxide. In this method, dibutyl terephthalate can be obtained with a yield between 80–85%. The original method of depolymerization of waste polyesters, including PET, was described in [48]. The essence of the method is the use of specific catalysts based on DBU and various imidazole derivatives (Figure 7).

The alcoholysis process was carried out in relatively mild conditions (70 °C) for 2 h. In the case of butanol, a PET conversion of 91% was obtained with a DBT yield of 73%, but for alcohols C5–C6 the yield of the corresponding terephthalic acid ester decreases.

For higher alcohols, clearly higher rates of yield of the appropriate diesters of terephthalic acid can only be obtained in processes carried out in more conventional parameters, i.e., higher temperature and pressure. According to the patent description CN102234227, the PET depolymerization process in the presence of higher alcohols can be carried out at a temperature of 180–200 °C for 4–9 h [98]. Zinc acetate is an effective catalyst for the reaction, but the described method does not provide detailed information on the conversion of PET and the yield of the corresponding terephthalic acid esters. Two other patents CN105503605 and CN106986326 describe the method of PET depolymerization with monohydric alcohols C4-C8 catalyzed by tetrabutyl titanate [99,100]. In addition, this method requires a slightly higher temperature range between 200–230 °C. However, the authors of both solutions also do not provide PET conversion rates and yields of the corresponding esters (Table 11).

#### 2.2.5. Alcoholysis with C8 Alcohols

Depolymerization of poly(ethylene terephthalate) can by conducted using higher alcohols, i.e., 2-ethylhexanole, isononyl and isodecyl alcohol. Products obtained in this manner are valuable chemicals that can be used as PVC plasticizers. This process can be catalyzed by organometallic catalysts, ionic liquids and superbases. One of the earliest works on alcoholysis of PET was presented by Gupta et al. [101]. Authors investigated the application of organotin catalyst in depolymerization of waste PET from different sources. The authors conducted depolymerization of waste PET from beverage and food bottles, packaging film, fabrics, car parts, photographic and X-ray films. Additionally, the catalyst was tested in depolymerization reaction of crystalized PET and glycol modified PET as well as PBT. Obtained DOTP samples were tested with PVC to determine their plasticizing properties. It was concluded that all obtained samples exhibit similar properties to commercial DOTP obtained from terephthalic acid. The most important difference was the difference in color of the product which is heavily influenced by the color of used substrate. Hardness, brittle point, tensile strength and elongation at break were at similar values regardless of the origin of applied plasticizer.

Ionic liquids

Ionic liquids are used as catalysts in alcoholysis of waste PET. A series of (3–sulfonic acid) propyltriethylammonium chloroironinate [HO_3_S–(CH_2_)_3_–NEt_3_]Cl–FeCl_3_ ionic liquids obtained with different molar fractions of FeCl_3_ as well as zinc and copper chloride derivatives were tested in depolymerization of waste polyethylene terephthalate with 2-ethylhexanol and their activity was compared to common transesterification catalysts like zinc acetate, zinc chloride and tetrabuthyl titaniate [102]. Reaction was carried out at temperature of 210 °C for the duration of 8 h using 2-ethylhexanol excess of 3.39. Analysis of postreaction mixture has shown that the highest yield of dioctyl terephthalate was obtained while using ionic liquids containing 0.67 and 0.75 molar fraction of FeCl_3_ and 0.67 molar fraction of ZnCl_2_. [HO_3_S–(CH_2_)_3_–NEt_3_]Cl–FeCl_3_ ionic liquid was reused seven times with only a small decrease in dioctyl terephthalate yield. [HO_3_S–(CH_2_)_3_–NEt_3_]Cl–ZnCl_2_ achieved also very good yields of dioctyl terephthalate (95.7%wt) while [HO_3_S–(CH_2_)_3_–NEt_3_]Cl–CuCl_2_ yielded only 37.3%wt of DOTP. Titanium butoxide, which is commonly used as a catalyst in synthesis of DOTP from terephthalic acid and 2-ethylhexanol, gave DOTP yield of 89.0%wt which is lower when compared to active ionic liquid catalysts. Similar results were obtained when ZnCl_2_ and Zn(CH_3_COO)_2_ were applied. Sulfuric acid has shown poor activity in depolymerization reaction with DOTP yield of 67.5 and PET conversion of 70.1%wt. This result is to be expected since PET depolymerization is a transesterification reaction and Bronsted acids are usually less active compounds in this reaction. Tested ionic liquid catalysts have largely shown good activity; however, a high catalyst concentration of catalyst was used in the reaction. Ionic liquids also plays a role as a cosolvent in PET alcoholysis [103]. A series of methylimidazolium ionic liquids containing different anions (Cl, Br, NO3, etc.) were tested in PET degradation trials in the presence of 2-ethylhexanole. Propylmethylimidazolium ([Amim]Cl) and butylmethylimidazolium ([Bmim]Cl) chlorides have proven to be the most effective cosolvents achieving PET degradation rates of 56.1 and 57.3%wt and DOTP yields of 42.4 and 43.2%wt, respectively. The process was conducted using 2:2:2 IL:2-Eh:PET weight ratios. Catalyst activity of titanium butoxide (TnBt) and zinc acetate was tested using [Bmim]Cl as cosolvent. [Bmim]Cl was reused four times with only marginal changes in PET degradation and DOTP yield. Application of ionic liquid cosolvent has significant impact on degradation of PET as well as on obtained yields of DOTP.

Deep eutectic solvents

Choline chloride deep eutectic solvents (DES) are also used in PET alcoholysis. Zhou et al. [104] investigated the use of DES comprised of choline chloride and various metal salts in PET depolymerization in the presence of 2-ethylhexanole (Table 12). Their activity was compared with other transesterification catalysts such as zinc, manganese and cobalt acetates among others. DES that were applied in depolymerization reaction showed high catalytic activity which resulted in high conversion rates of PET and good DOTP yields. ChCl/Za(Ac)_2_ has shown the best results achieving conversion close to 100%wt and 84.2%wt yield of DOTP. The depolymerization process was conducted at 185 °C for 60 min with 2-Eh/PET ratio of 3.4 and DES concentration of 5%wt regarding initial PET mass. Choline chloride deep eutectic solvents have shown very good catalytic activity when compared to standard transesterification catalysts such as titanium butoxide or zinc acetate, which under the same reaction conditions yielded lower conversion rates of 78.5%wt and 46.4%wt, respectively, and DOTP yields (63.8%wt and 40.6%wt).

## 3. Conclusions

The significant amount of waste generated by poly(ethylene terephthalate) requires the development of a recycling process chain in which chemical recycling plays an important role. On the one hand, it allows the depolymerization of degraded plastics that do not meet the quality requirements to be used in mechanical recycling, and on the other hand, provides an opportunity to process cheap waste and obtain products with greater added value. It can be widely used in the recycling of both packaging plastics and textiles, or other waste generated with PET.

Chemical depolymerization processes of poly(ethylene terephthalate) can be carried out in several ways, using the susceptibility of the ester group to select chemical reactions such as hydrolysis and transesterification. The products obtained as a result can be divided into two groups, the first of which includes compounds obtained by hydrolysis, methanolysis, ethanolysis and glycolysis used in the synthesis of polymers. The second group includes terephthalic esters with plasticizing properties of plastics obtained by depolymerization with higher monohydric alcohols.

The use of hydrolysis allows terephthalic acid to be obtained, which after purification can be reused in the polymerization reaction to obtain plastics. This process is relatively widely described in the scientific literature and has found practical application in industry. It requires the use of high temperatures and pressures and, in the case of alkaline hydrolysis, a significant number of bases. Inorganic hydroxides, inorganic acids, organic acids and heteropolyacids, as well as “superacids” and quaternary ammonium salts are used as catalysts in the process. However, it allows for high PET conversion rates and yields in excess of 90%. An important problem in the use of this method is the purification of crude terephthalic acid to the purity required by polymer manufacturers.

The alcoholysis process includes PET depolymerization using monohydric alcohols such as methanol, ethanol, butanol and higher alcohols such as butanol or 2-ethylhexanol. The methanolysis reaction is another chemical recycling process that allows raw materials to be obtained that can be directly used in the synthesis of a new polymer. The process usually requires high temperatures, which due to the low boiling point of methanol necessitates the use of a high-pressure environment. Typical catalysts used in this process are metal acetates, with zinc acetate exhibiting the highest activity. Other salts, “superbases” and ionic liquids are also used. The process can also be carried out in supercritical conditions, which ensures high efficiency without the need for a catalyst. Polyethylene terephthalate ethanolysis is seen as an alternative to the methanolysis process to eliminate harmful methyl alcohol. The process is carried out analogously to methanolysis at elevated temperature and pressure using analogous catalysts. The process can also be carried out under supercritical conditions. Glycolysis is one of the most widely described PET depolymerization processes in the scientific literature. The presented results cover a wide range of catalysts used from typical compounds such as metal salts, metal oxides, metal complexes, polyoxometalates to ionic liquids.

Higher alcohols such as butanol or 2-ethylhexanol can be used in the synthesis of terephthalic plasticizers using catalysts such as superbases, ionic liquids, Lewis acids, acetates and deep eutectic solvents. Due to the high boiling point of the alcohols used, depolymerization processes do not require the use of high pressures. The obtained products have plasticizing properties similar to plasticizers obtained by traditional methods. They are potential substitutes for plasticizers obtained by traditional methods.

The re-use of waste poly(ethylene terephthalate) is an increasingly common practice, mainly due to legal regulations requiring producers to introduce more waste raw material into production stream. Recycling of waste PET by introducing waste substrate in to production leads to gradual degradation of the processed material, which forces the development of effective methods of its chemical recycling. Chemical recycling of PET can be divided into processes that result in compounds used in the production of fresh poly(ethylene terephthalate) and upcycling processes, which result in products used in other fields, with higher added value. Due to their high potential for practical application, it is expected that their future development will focus on developing methods to achieve high conversions and yields in order to obtain a high-quality product. In particular, the future development of PET chemical recycling methods will focus on the development of catalysts that will achieve high product performance with low-mass feedstock ratios. The use of superacids in hydrolysis and acetates, nanodispersion and processes under supercritical conditions in methanolysis and acetates in the glycolysis process seems to be a future direction. Chemical recycling towards value-added products represents an interesting path for waste PET treatment. From the group of products that can be obtained by PET alcoholization, the most promising are the terephthalates of alcohols C_8_ to C_10_. These compounds are widely used as PVC plasticizers, which are traditionally obtained from terephthalic acid and can be seen as a viable alternative for traditional process.

By characterizing the depolymerization methods and catalysts described in the scientific literature, this review will contribute to a better understanding of the state of knowledge about the chemical recycling of poly(ethylene terephthalate). It will also undoubtedly be of help to scientists planning to start research on this topic.

## Figures and Tables

**Figure 1 molecules-28-06385-f001:**
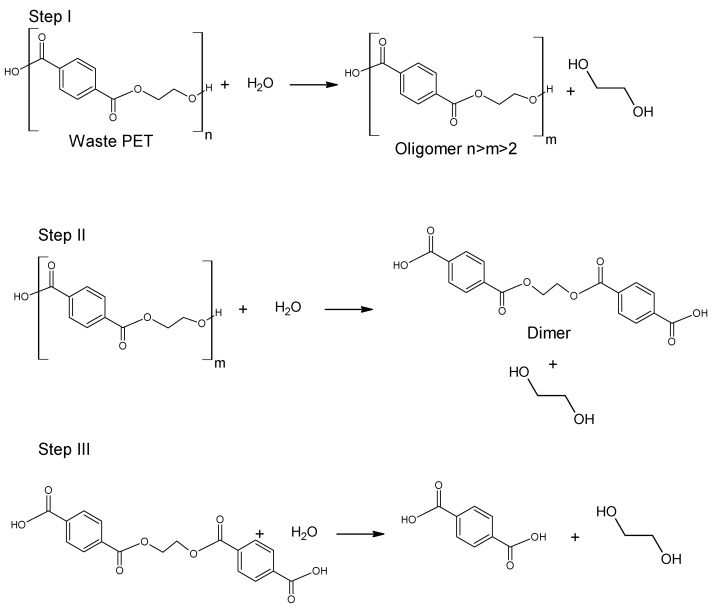
Hydrolysis of poly(ethylene terephthalate).

**Figure 2 molecules-28-06385-f002:**
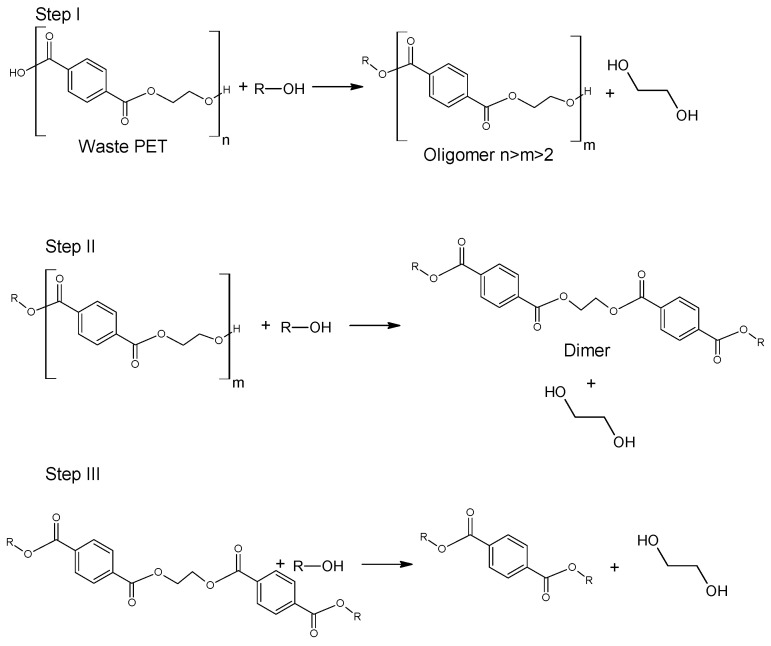
Alcoholysis of poly(ethylene terephthalate).

**Figure 3 molecules-28-06385-f003:**
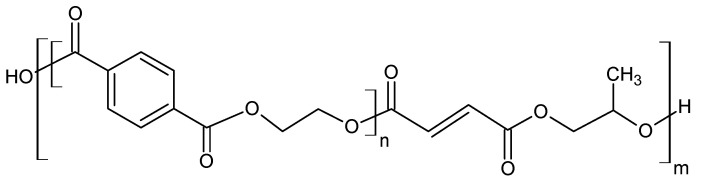
Structure of Thixotropic and Chemoresistant Unsaturated Polyester Resins [3].

**Figure 4 molecules-28-06385-f004:**
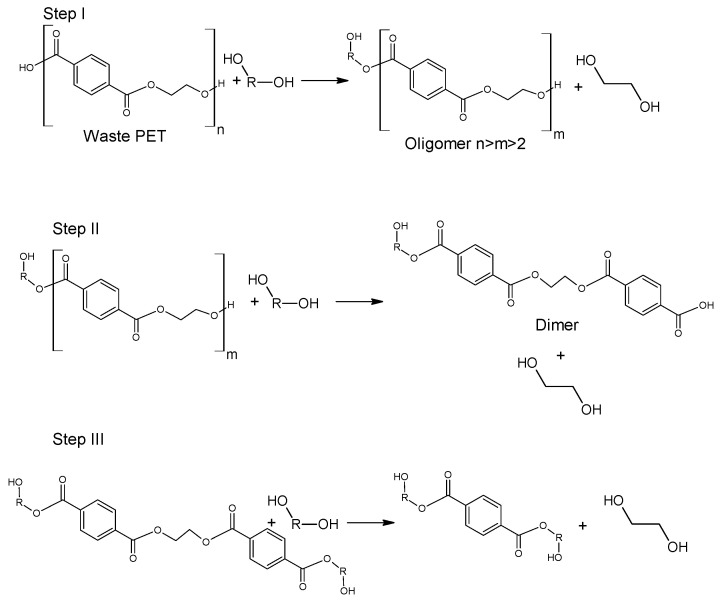
Glycolysis of poly(ethylene terephthalate) [57].

**Figure 5 molecules-28-06385-f005:**
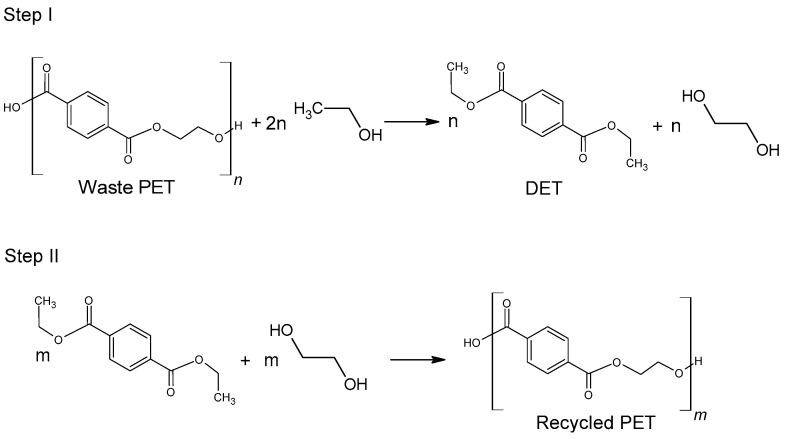
Recycling of waste PET via direct ethanolysis.

**Figure 6 molecules-28-06385-f006:**
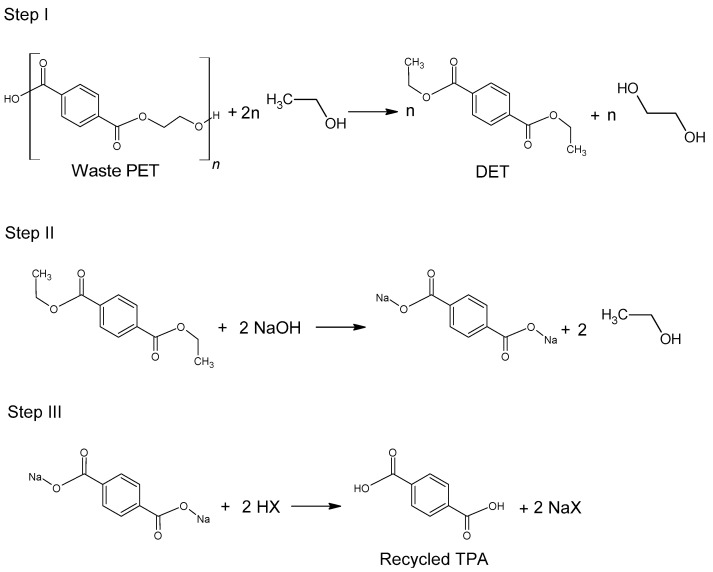
Three-step recycling process of waste PET into terephthalic acid.

**Figure 7 molecules-28-06385-f007:**
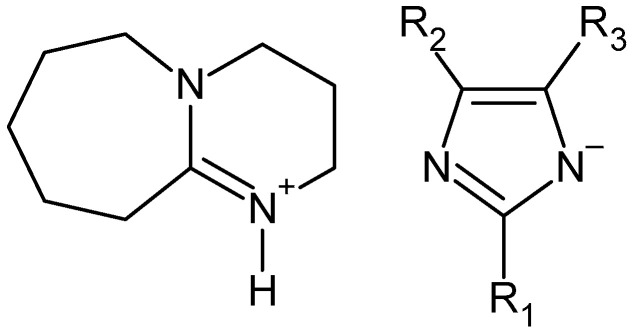
DBU ionic liquid as catalyst for alcoholysis of waste polyesters (PC, PLA, PHB, and PET) [48].

**Table 1 molecules-28-06385-t001:** PET hydrolysis in alkaline process conditions.

TPA Yield [%]	Reaction Temperature [°C]	Reaction Time [min]	Pressure [bar]	Catalyst	Solvent	Reference
97	120–200	60	n/a	NaOH	Water	[12]
92	200	25	n/a	NaOH	Water	[20]
95	220	n/a	26	NaOH	Water	[23]
95	80	20	1	NaOH	Water/Ethanol	[24]
99	80	20	1	NaOH	Water	[25]
95	n/a ^1^	60	1	NaOH + TBAJ	Water	[26] ^1^
99	110	300	n/a	NaOH + [CTA]_3_PW	Water	[27] ^2^
93	145	120	n/a	NaOH + [CTA]_3_PW	Water	[28] ^2^

^1^ under microwave irradiation, ^2^ cetyltrimethylammonium phosphotungstate, n/a—not applicable.

**Table 2 molecules-28-06385-t002:** PET hydrolysis in acidic process conditions.

TPA Yield [%]	Reaction Temperature [°C]	Reaction Time [min]	Pressure [bar]	Catalyst	Solvent	Reference
75	180	180	n/a	H_2_SO_4_	Water	[28]
73	180	180	n/a	H_3_PW_12_O_40_	Water	[28]
93	180	180	n/a	[CTA]_3_PW	Water	[28]
96.2	150	90	n/a	PTSA	Water	[29]
99.2	160	12 h	150	SO_4_^2−^/TiO_2_	Water	[30] ^1^
99.5	160	15 h	150	WO_3_/SiO_2_	Water	[31] ^1^
95.5	220	180	n/a	TPA	Water	[32]

Note ^1^ under supercritical CO_2_, n/a—not applicable.

**Table 3 molecules-28-06385-t003:** Methanolysis reaction parameters.

Yield [%]	Alcohol/Polymer Ratio	Reaction Temperature [°C]	Reaction Time [min]	Pressure [bar]	Catalyst	Catalyst Amount [%wt.]	Reference
98	46.2	160	20	n/a	Zn(OAc)_2_	1	[37]
97.76	2.38	130	120	8.5	Zn(OAc)_2_/Pb(OAc)_2_	5.1	[38]
0.0	2.16	64.7	900	1	Zn(OAc)_2_	0.07	[37]
93.1	50	25	1440	1	K_2_CO_3_	0.2 ^a^	[39]
89.3	50	25	1440	1	TBD	0.2 ^a^	[39]
85.5	50	25	1440	1	CH_3_OK	0.2 ^a^	[39]
95	5	200	45	n/a	Na_2_SiO_3_	5	[40]
74	200	180	150	n/a	MgP	3	[41]
78	7.5	200	120	n/a	BLA	20.8	[42]
97	6	170	15	n/a	ZnO nanodispersion	3.5	[43]
78	5	170	240	n/a	[Bmim]_2_[CoCl_4_]	1.6	[47]
75	5	140	180	n/a	[HDBU][Im]	1.6	[47]
99.79	6	298	112	n/a	-	-	[48]

^a^ % of PET degradation; n/a—not applicable.

**Table 4 molecules-28-06385-t004:** Results of PET glycolysis [62].

Catalyst	Catalyst Concentration	Reaction Temperature	Reaction Time [h]	PET Conversion [%]	BHET Yield [%]
-	-	-	-	0	0
Ni(NO_3_)_2_ × 6H_2_O	1 [%wt.]	190 °C	3	13	4
FeSO_4_ × 7H_2_O	1 [%wt.]	190 °C	3	44	12
CuSO_4_ × 2H_2_O	1 [%wt.]	190 °C	3	49	17
CuCl_2_ × 2H_2_O	1 [%wt.]	190 °C	3	53	19
Zn(OAc)_2_ × 2H_2_O	1 [%wt.]	190 °C	3	55	23
FeCl_3_ × 6H_2_O	1 [%wt.]	190 °C	3	57	42
AlCl_3_	1 [%wt.]	190 °C	3	59	44
ZnCl_2_	1 [%wt.]	190 °C	3	60	45
MnCl_2_ × 4H_2_O	1 [%wt.]	190 °C	3	62	48
NiSO_4_ × 7H_2_O	1 [%wt.]	190 °C	3	100	55
NiCl_2_ × 6H_2_O	1 [%wt.]	190 °C	3	100	55
ZnSO_4_ × 7H_2_O	1 [%wt.]	190 °C	3	100	57
CoCl_2_ × 6H_2_O	1 [%wt.]	190 °C	3	100	65
ZrCl_4_	1 [%wt.]	190 °C	3	100	69
BEt_3_	1 [%wt.]	190 °C	3	100	56
CoCl_2_ 1:3 dcype	1.5 [mol % of CoCl_2_]	190 °C	3	-	10
CoCl_2_ 1:1 dcype	1.5 [mol % of CoCl_2_]	190 °C	1	-	58
[Co(dcype)Cl_2_]	1.5 [mol % of CoCl_2_]	170 °C	3	-	10
[Co(dcype)Cl_2_]	1.5 [mol % of CoCl_2_]	190 °C	1	-	58
[Co(dcype)Cl_2_]	1.5 [mol % of CoCl_2_]	190 °C	2	-	75
[Co(dcype)Cl_2_]	1.5 [mol % of CoCl_2_]	190 °C	3	-	75
[Co(dcype)Cl_2_]	3 [mol % of CoCl_2_]	190 °C	3	-	82
Ni(COD)_2_	1 [%wt. of Ni(COD)_2_]	190 °C		100	55
[Ni(COD)_2_] 1:4 PPh_3_	1 [%wt. of Ni(COD)_2_]	190 °C	3	27	13
[Ni(COD)_2_] 1:4 P(OiPr)_3_	1 [%wt. of Ni(COD)_2_]	190 °C	3	0	0
[Ni(COD)_2_] 1:2 dcype	1 [%wt. of Ni(COD)_2_]	190 °C	3	100	59
[Ni(COD)_2_] 1:2 dppe	1 [%wt. of Ni(COD)_2_]	190 °C	3	100	67
[Ni(COD)_2_] 1:2 dppf	1 [%wt. of Ni(COD)_2_]	190 °C	3	30	15
[(COD)Ni(dppe)]	1 [%wt. of Ni(COD)_2_]	190 °C	3	-	44
[Ni(COD)_2_] 1:1 dppe	1 [%wt. of Ni(COD)_2_]	190 °C	3	-	48
[Ni(COD)_2_] 1:2 dppe	1 [%wt. of Ni(COD)_2_]	190 °C	1	-	30
[Ni(COD)_2_] 1:2 dppe	1 [%wt. of Ni(COD)_2_]	190 °C	2	-	57
[Ni(COD)_2_] 1:2 dppe	1 [%wt. of Ni(COD)_2_]	190 °C	3	-	67
[Ni(COD)_2_] 1:2 dppe	1 [%wt. of Ni(COD)_2_]	190 °C	5	-	59
[Ni(COD)_2_] 1:2 dppe	1 [%wt. of Ni(COD)_2_]	200 °C	3	-	71

Note: Pressure = 1 [atm]; Glycol:PET ratio = 10 mL:1 g.

**Table 5 molecules-28-06385-t005:** Results of PET glycolysis with EG and DMSO with 0.25 [g] of a catalyst [63].

Catalyst	Time [min]	Yield of BHET with Solvent [%]	BHET Yield without Solvent [%]
Zn(OAc)_2_ × 2H_2_O	5	83.88 ± 1.12	42.98 ± 1.76
Zn(OAc)_2_ × 2H_2_O	1	82.97 ± 2.44	20.11 ± 2.01
Zn(NO_3_)_2_ × 6H_2_O	5	78.64 ± 2.60	25.69 ± 2.37
Zn(NO_3_)_2_ × 2H_2_O	1	53.48 ± 2.46	7.32 ± 2.33
ZnSO_4_ × 7H_2_O	5	57.21 ± 2.14	1.27 ± 0.29
Co(OAc)_2_ × 4H_2_O	5	78.69 ± 1.87	24.73 ± 1.88
Ni(OAc)_2_ × 4H_2_O	5	21.65 ± 1.82	1.22 ± 0.20
Cu(OAc)_2_ × H_2_O	5	14.68 ± 1.01	1.16 ± 0.32
Mn(OAc)_2_ × 4H_2_O	5	80.77 ± 1.70	42.15 ± 2.37
Urea/Zn(OAc)_2_	5	77.90 ± 2.28	48.17 ± 1.51
K_6_SiW_11_ZnO_39_(H_2_O)	5	65.94 ± 2.86	38.77 ± 2.62
[Bmim] Zn(OAc)_3_	5	72.44 ± 2.29	38.40 ± 2.53

Note: Pressure = 1 [atm]; Glycol:PET wt. ratio = 6:1 without solvent; Glycol:PET wt. ratio = 2:1 with solvent; Temperature = 190 °C.

**Table 6 molecules-28-06385-t006:** Results of the PET glycolysis in the most efficient parameters [65].

Catalyst	Temperature [°C]	PET Conversion [%]	BHET Yield [%]
CoFe_2_O_4_	195	91.2	72.6
CoFe_2_O_4_/[C_6_COOHbim]Br	195	95.3	79.7
CoFe_2_O_4_/[C_10_COOHbim]Br	195	98.4	88.03
CoFe_2_O_4_/[C_10_COOHbim]NTf_2_	195	99.6	86.6
CoFe_2_O_4_/[C_10_COOHbim]OAc	195	100	95.4
[C_6_COOHbim]Br	205	19.8	8.5
[C_10_COOHbim]Br	205	22.9	9.5
[C_10_COOHbim]NTf_2_	205	20.4	7.5
[C_10_COOHbim]OAc	205	60.5	32.1

Note: CoFe_2_O_4_—MNP; bim—butyl imidazole; NTf_2_—bis[(trifluoromethyl) sulfonyl] imide; OAc—CH_3_COO^−^; C_6_COOH—hexanoic acid; C_10_COOH—decanoic acid; Pressure = 1 [atm]; Glycol:PET wt. ratio = 5:1; Duration 2.5 h.

**Table 7 molecules-28-06385-t007:** Results of PET glycolysis reactions with POMs catalysts [67].

Catalyst	Reaction Time [min]	PTE Conversion [%]	BHET Yield [%]
Na_12_[WZn_3_(H_2_O)_2_(ZnW_9_O_34_)_2_]	40	100.00	84.48
Na_12_[WZnCo_2_(H_2_O)_2_(ZnW_9_O_34_)_2_]	50	100.00	83.95
Na_12_[WZnCo_2_(H_2_O)_2_(ZnW_9_O_34_)_2_]	40	99.83	84.61
Na_12_[WZnNi_2_(H_2_O)_2_(ZnW_9_O_34_)_2_]	60	100.00	83.96
Na_12_[WZnNi_2_(H_2_O)_2_(ZnW_9_O_34_)_2_]	40	98.13	77.48
Na_12_[WZnCu_2_(H_2_O)_2_(ZnW_9_O_34_)_2_]	65	100.00	83.61
Na_12_[WZnCu_2_(H_2_O)_2_(ZnW_9_O_34_)_2_]	40	99.18	78.42
Na_12_[WZnMn_2_(H_2_O)_2_(ZnW_9_O_34_)_2_]	44	100.00	84.11
Na_12_[WZnMn_2_(H_2_O)_2_(ZnW_9_O_34_)_2_]	40	99.90	82.88
Na_12_[WZn_3_(H_2_O)_2_(CoW_9_O_34_)_2_]	110	100.00	82.42
Na_12_[WZn_3_(H_2_O)_2_(CoW_9_O_34_)_2_]	40	66.90	69.88

Note: Pressure = 1 [atm]; Glycol:PET wt. ratio = 4:1; Temperature = 190 °C.

**Table 8 molecules-28-06385-t008:** Results of PET glycolysis reactions with OPA catalyst [70].

Catalyst Amount [mg]	EG Amount [g]	Time [h]	PET Conversion [%]	BHET Yield [%]
20	2	1	100	63
30	2	1	100	71
50	2	1	100	75
70	2	1	100	69
100	2	1	100	71
50	1	1	100	61
50	1.5	1	100	73
50	2.5	1	100	78
50	3	1	100	76
50	2.5	0.5	80	54
50	2.5	1.5	100	79
50	2.5	2	100	76
50	2.5	2.5	100	71

Note: Pressure = 1 [atm]; PET amount = 480 mg; Temperature = 190 °C.

**Table 9 molecules-28-06385-t009:** Results of PET glycolysis reactions with LAIL’s catalysts [71].

Molar Ratios [Hmim]Cl:ZnCl_2_:CoCl_2_:FeCl_3_:CuCl_2_	PET Conversion [%]	BHET Yield [%]
1:1:0:0:0	100	76.8
1:0:1:0:0	85.8	38.1
1:0.25:0.75:0:0	100	75.2
1:0.5:0.5:0:0	100	87.1
1:0.75:0.25:0:0	100	79.6
1:0:0:1:0	99.5	51.5
1:0.25:0:0.75:0	100	60.8
1:0.5:0:0.5:0	98	55.1
1:0.75:0:0.25:0	99.8	37.5
1:0:0:0:1	4.8	–
1:0.25:0:0:0.75	14.5	–
1:0.5:0:0:0.5	99	73.0
1:0.75:0:0:0.25	100	76.8

Note: [Hmim]Cl:ZnCl_2_—1:1 = [Hmim]ZnCl_3_; [Hmim]Cl:ZnCl_2_:CoCl_2_—1:0.5:0.5 = [Hmim]ZnCl_3_:[Hmim]CoCl_3_—1:1; Pressure = 1 [atm]; Glycol:PET wt. ratio = 11:1; Duration = 2 h; Temperature = 190 °C.

**Table 10 molecules-28-06385-t010:** PET alcoholysis with ethanol.

DET Yield [%]	Reaction Temperature [°C]	Reaction Time [min]	Pressure [bar]	Catalyst	Solvent	Reference
97.3	200	210	n/a	Zn(OAc)_2_	Ethanol	[82]
92	220	120	n/a	Zn(OAc)_2_	Ethanol	[83]
97	180	60	n/a	Zn(OAc)_2_ + Al(OH)_3_	Ethanol	[84]
95.8	80	20	1	Sulphonic IL	Ethanol	[85]
98.5	240	300	115–165 ^1^	-	Ethanol	[94]
>99	255	90	115 ^1^	Co_3_O_4_ or NiO	Ethanol	[89]
92.2	270	60	n/a ^1^	ZnO/Al_2_O_3_	Ethanol	[90]
98	255	45	115	[bmim]BF_4_	Ethanol	[91,92]
95	275–300	90	n/a ^1^	-	Ethanol	[93]
98	310	60	n/a ^1^	-	Ethanol	[94]

^1^ under supercritical conditions, n/a—not applicable.

**Table 11 molecules-28-06385-t011:** PET alcoholysis with alcohols C3–C7.

DET Yield [%]	Alcohol	Reaction Temperature [°C]	Reaction Time [min]	Catalyst	Reference
92.3	1-Butanol	190	300	Sulphonic IL	[85]
95.3	1-Hexanol	190	600	Sulphonic IL	[85]
80–85	1-Butanol	50–70	90–180	TBD(DBU) + CH_3_ONa	[97]
78	1-Propanol	70	120	[HDBU][Im]	[48]
73	1-Butanol	70	120	[HDBU][Im]	[48]
n/a	C4–C8	180–200	240–540	Zn(OAc)_2_	[98]
n/a	C4–C8	200–230	120–600	TBT	[99]
n/a	C4–C8	200–230	120–600	TBT	[100]

**Table 12 molecules-28-06385-t012:** Results of PET alcoholysis with 2-ethylhexanole.

Catalyst	Catalyst Concentration [%wt.]	Reaction Temperature [°C]	Reaction Time [min]	Conversion [%]	Yield [%]	Ref.
ZnCl_2_	20	210	480	93.2	87.6	[102]
Zn(CH_3_COO)_2_	20	210	480	92.5	88.2	[102]
(CH_3_CH_2_CH_2_CH_2_O)Ti	20	210	480	93.2	89.0	[102]
H_2_SO_4_	20	210	480	70.1	67.5	[102]
[HO_3_S–(CH_2_)_3_–NEt_3_]Cl	20	210	480	40.1	35.4	[102]
[C_4_H_6_N_2_(CH_2_)_3_SO_3_H]_3_PW_12_O_40_	20	210	480	97.5	94.7	[102]
[HO_3_S–(CH_2_)_3_–NEt_3_]–FeCl_3_ (x = 0.50)	20	210	480	42.4	38.6	[102]
[HO_3_S–(CH_2_)_3_–NEt_3_]Cl–FeCl_3_ (x = 0.60)	20	210	480	86.0	84.2	[102]
[HO_3_S–(CH_2_)_3_–NEt_3_]Cl–FeCl_3_ (x = 0.64)	20	210	480	94.2	88.6	[102]
[HO_3_S–(CH_2_)_3_–NEt_3_]Cl–FeCl_3_ (x = 0.67)	20	210	480	100	97.6	[102]
[HO_3_S–(CH_2_)_3_–NEt_3_]Cl–FeCl_3_ (x = 0.75)	20	210	480	100	97.9	[102]
[HO_3_S–(CH_2_)_3_–NEt_3_]Cl–ZnCl_2_ (x = 0.67)	20	210	480	100	95.7	[102]
[HO_3_S–(CH_2_)_3_–NEt_3_]Cl–FeCl_2_ (x = 0.67)	20	210	480	87.7	78.9	[102]
[HO_3_S–(CH_2_)_3_–NEt_3_]Cl–CuCl_2_ (x = 0.67)	20	210	480	40.3	37.3	[102]
[C_4_mim]Cl–FeCl_3_ (x = 0.67)	20	210	480	92.1	91.4	[102]
-	-	190	240	1.7	1.2	[103]
[Amim]Cl ^b^	-	190	240	56.1 ^a^	42.4	[103]
[Bmim]Cl ^b^	-	190	240	57.3 ^a^	43.4	[103]
[Bmim]Br ^b^	-	190	240	46.2 ^a^	37.3	[103]
[Bmim]NO_3_ ^b^	-	190	240	10.5 ^a^	6.5	[103]
[Hmim]CF_3_SO_3_ ^b^	-	190	240	5.2 ^a^	3.7	[103]
[Bmim]HSO_4_ ^b^	-	190	240	28.6 ^a^	20.5	[103]
[Bmim]BF_4_ ^b^	-	190	240	4.3 ^a^	3.1	[103]
B[mim]PF_6_ ^b^	-	190	240	3.7 ^a^	2.2	[103]
Ti(OC_4_H_9_)_4_ ^c^	1.2	190	240	98.1	86.7	[103]
Zn(CH_3_COO)_2_ ^c^	1.2	190	240	97.5	85.9	[103]
ChCl	5	185	60	3.4	-	[104]
ChCl/Zn(Ac)_2_ (1:1)	5	185	60	100	84.2	[104]
ChCl/Mn(Ac)_2_ (1:1)	5	185	60	96.5	80.6	[104]
ChCl/Co(Ac)_2_ (1:1)	5	185	60	90.4	74.1	[104]
ChCl/Cu(Ac)_2_ (1:1)	5	185	60	91.7	76.2	[104]
ChCl/FeCl_3_ (1:1)	5	185	60	94.5	78.3	[104]
Zn(Ac)_2_	5	185	60	46.4	40.6	[104]
Mn(Ac)_2_	5	185	60	44.2	36.6	[104]
Co(Ac)_2_	5	185	60	32.9	27.1	[104]
Cu(Ac)_2_	5	185	60	33.8	27.6	[104]
FeCl_3_	5	185	60	43.8	35.7	[104]
CoCl_2_	5	185	60	31.6	26.5	[104]
H_2_SO_4_	5	185	60	41.4	34.5	[104]
Ti(OC_4_H_9_)_4_	5	185	60	78.5		[104]

Note. ^a^ % of PET degradation, ^b^ cosolvent, ^c^ [Bmim]Cl as cosolvent.

## Data Availability

No new data was created.

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
