# Peer review of "Comparsion of Catalyst Effectiveness in Different Chemical Depolymerization Methods of Poly(ethylene terephthalate)"

_molecules, 2023, doi:10.3390/molecules28176385_

Round 1
Reviewer 1 Report
This paper presents an overview of the methods of chemical recycling of polyethylene terephthalate (PET) described in the scientific literature in recent years. The review was focused on methods of chemical recycling of PET including hydrolysis and broadly understood alcoholysis of polymer ester bonds including methanolosis, ethanolosis, glycolysis and reactions with higher alcohols. However, the future development of the chemical recycling of PET should be given.
Minor editing of English language required
Reviewer 2 Report
Tha material is clearly organized and presented. Several Tables help the readers to have a clear view.
The paper deserves publication, but improvement is necessary after careful rereading, considering the following suggestions.
TABLE 1: notes at the end of the table are numbered (1 under microwave irradiation, 2 cetyltrimethylammonium phosphotungstate) , but corresponding numbers are not present in the table
The same in TABLE 2
line 123 the prefix para-, as in p-toluene... must be in italics p-toluene...
line 192 "... in low temperatures in methanolysis of ,,,", better "at low temperatures..."
line 230 Interestingly must be followed by a comma (Interestingly, ...)
line 225 SiO2 2 is not subscript (SiO2)
line 282 "Except of being widely discussed..." I am not sure of the meaning. Maybe "besides being widely discussed ..."
line 304 "... The highest productivity were..." Either " The highest productivity was.." or " The highest productivities were..."
line 382 "ration" according to Oxford Dictionary is "a fixed amount of a commodity officially allowed to each person during a time of shortage, as in wartime."
The correct word here should be "ratio", although I'd rather say "... the ratio between the amount of water soluble monomer (WSM) versus the theoretical amount of WSM
line 384 "Presented results exhibits ..." results is plural "Presented results exhibit"
line 426 "...indicate, that from tested catalysts..." punctuation is not correct. it should be "indicate that, from tested catalysts, ..."
Line 559 "Others" ??? do you mean "Other systems"?
line 762 NO3 3 is not subscript (NO3)
the quality is good. Only minor changes have been suggested 8see general comments)
